# Neural Circuit Architectural Priors
# for Embodied Control

**Nikhil X. Bhattasali, Anthony M. Zador, Tatiana A. Engel**
NeuroAI Program, Cold Spring Harbor Laboratory
{bhattas,zador,engel}@cshl.edu

## Abstract

Artificial neural networks for motor control usually adopt generic architectures like fully connected MLPs. While general, these *tabula rasa* architectures rely on large amounts of experience to learn, are not easily transferable to new bodies, and have internal dynamics that are difficult to interpret. In nature, animals are born with highly structured connectivity in their nervous systems shaped by evolution; this innate circuitry acts synergistically with learning mechanisms to provide inductive biases that enable most animals to function well soon after birth and learn efficiently. Convolutional networks inspired by visual circuitry have encoded useful biases for vision. However, it is unknown the extent to which ANN architectures inspired by neural circuitry can yield useful biases for other AI domains. In this work, we ask what advantages biologically inspired ANN architecture can provide in the domain of motor control. Specifically, we translate *C. elegans* locomotion circuits into an ANN model controlling a simulated Swimmer agent. On a locomotion task, our architecture achieves good initial performance and asymptotic performance comparable with MLPs, while dramatically improving data efficiency and requiring orders of magnitude fewer parameters. Our architecture is interpretable and transfers to new body designs. An ablation analysis shows that constrained excitation/inhibition is crucial for learning, while weight initialization contributes to good initial performance. Our work demonstrates several advantages of biologically inspired ANN architecture and encourages future work in more complex embodied control.

## 1 Introduction

Artificial neural networks (ANNs) for motor control usually adopt generic architectures like fully connected multi-layered perceptrons (MLPs) [Pierson and Gashler, 2017, Levine et al., 2016, Bin Peng et al., 2020, Heess et al., 2016]. While general, these *tabula rasa* architectures rely on large amounts of experience to learn. Data efficiency is especially desirable because, unlike computer vision and natural language processing which have benefited greatly from available large datasets [Bommasani et al., 2021], motor control requires interacting with an environment to gather experience, which is time-consuming, laborious, and possibly unsafe [Kroemer et al., 2020]. Transfer is also a challenge, as experience is usually body-specific, and an ANN trained to control one body is not easily adapted to new bodies. In addition, architectures like MLPs are difficult to interpret, as their internal dynamics are distributed across units and non-trivial to relate to agent behavior [Merel et al., 2019a].

In nature, animals are born with highly structured connectivity in their brains and nervous systems that has been shaped over millennia by evolution [Zador, 2019]. In some cases, this innate circuitry confers abilities with little or no learning; in others, it guides the learning process by providing strong inductive biases [Lake et al., 2017]. These innate and learning mechanisms act synergistically, enabling most animals to function well soon after birth, while continuing to acquire and improve skills efficiently, e.g. a horse learning to walk with only a couple hours of experience. Moreover, despite

36th Conference on Neural Information Processing Systems (NeurIPS 2022).

species-specific variations, there is a significant amount of shared architecture (e.g. cerebellum, basal ganglia) and design principles (e.g. hierarchical modularity, partial autonomy) even between distantly related species [Merel et al., 2019b]. The connectivity of neural circuits is often highly structured and sparse [Luo, 2021], in sharp contrast to the all-to-all connectivity of MLPs. Moreover, evolution has progressively built more advanced abilities on lower-level circuits, leveraging modular structure to transfer existing working designs to new animal bodies [Cisek, 2019, Merel et al., 2019b]. Taken together, these findings from neuroscience suggest that structured neural circuits in animals instantiate efficient, transferable, and modular solutions for high-dimensional embodied control.

Capturing some of this structure in model architecture may enable ANNs to narrow the gap between artificial and natural systems. For example, convolutional networks were inspired by biological visual circuitry, and the inductive biases of spatial locality and weight sharing that they encode have yielded substantial improvements in performance, data efficiency, and parameter efficiency for vision tasks [Lindsay, 2021]. However, it is unknown the extent to which ANN architectures inspired by neural circuitry can yield useful inductive biases for other AI domains.

In this work, we ask what advantages biologically inspired ANN architecture can provide in the domain of motor control. Specifically, we translate *C. elegans* locomotion circuits into an ANN model controlling a simulated Swimmer agent selected from a standard AI benchmark [Tassa et al., 2020]. Our architecture is an instance of what we call a "Neural Circuit Architectural Prior" (NCAP), to denote an ANN architecture that encodes prior structure inspired by biological neural circuits.

On a locomotion task, our architecture achieves good initial performance and asymptotic performance comparable with MLPs, while dramatically improving data efficiency and requiring orders of magnitude fewer parameters. Our architecture is interpretable and transfers to new body designs. An ablation analysis shows that constrained excitation/inhibition is crucial for learning, while weight initialization contributes to good initial performance. Our work demonstrates several advantages of biologically inspired ANN architecture and encourages future work in more complex embodied control.

In summary, the primary contributions of this work are:

1. An ANN architecture inspired by *C. elegans* locomotion circuits that combines the discrete-time ANN formalism that is standard in machine learning with features from computational neuroscience like constraints on synapse sign (i.e. excitation vs. inhibition) and special cell types (i.e. intrinsic oscillators);

2. An evaluation of our model's initial performance, asymptotic performance, data efficiency, parameter efficiency, interpretability, and transfer compared to standard MLP architectures; and

3. An ablation analysis of the effects of weight sharing, sign constraints, initialization, and sparse connectivity on performance and learning.

Code and videos are available at: `https://sites.google.com/view/ncap-swimmer`

## 2   Related Work

**Movement Priors**   A motor controller ultimately generates joint-space torques $\tau$ to apply at each actuator. While a learning-based controller can directly generate torques [Levine et al., 2016] or world-space positions $x$ or accelerations $\ddot{x}$ that are transformed into torques [Khatib, 1987], movement priors are often adopted to introduce abstraction, incorporate prior knowledge, and improve performance and learning speed [Kroemer et al., 2020]. Movement priors can be usefully grouped into 3 categories:

(1) *Trajectory Priors* encode desired movement through analytic equations of motion, e.g. Dynamic Movement Primitives use parameterized differential equations to implement stable attractor dynamical systems expressing different motion shapes [Schaal, 2006, Pastor et al., 2009], and Policies Modulating Trajectory Generators augment hand-engineered trajectories with learned residual terms to enhance robustness and flexibility [Iscen et al., 2019]. Generally, trajectory priors are interpretable and work well when desired movement is adequately captured by analytic primitives (e.g. sinusoids, splines); however, they can require much time and manual effort to design robustly.

(2) *Behavioral Imitation Priors* encode desired movement by training parameterized functions like ANNs to imitate reference motions generated from motion capture or manual keyframing, e.g. Bin Peng et al. [2020] train expert policies to imitate animal motion capture, and Neural

Probabilistic Motor Primitives compress expert policies trained to imitate human motion capture into a common, reusable embedding space [Merel et al., 2019c]. Generally, behavioral imitation priors have reproduced diverse movements and trade the time/effort of designing equations with that of compiling reference motions from humans/animals and retargeting the motions to agent bodies.

(3) *Architectural Priors* encode desired movement indirectly through specialized ANN architectures that establish inductive biases, e.g. Heess et al. [2016] decompose an agent into a loosely neuro-inspired hierarchy consisting of a high-level "cortical network" with access to exteroceptive signals and a low-level "spinal network" with access to only proprioceptive signals, and Heess et al. [2017] extend this architecture to more challenging locomotion tasks. Generally, architectural priors can significantly improve performance and data efficiency over generic/monolithic architectures; however, architectural priors still represent an under-explored space in the AI motor control literature, and much existing work has only specified high-level architectural structure while retaining generic low-level connectivity, e.g. the above-referenced "cortical" and "spinal" networks both use dense MLP connectivity, rather than connectivity that resembles biological cortical and spinal circuitry.

In this work, we investigate the potential of architectural priors that more closely resemble neural circuitry. In doing so, we demonstrate one path for how prior knowledge can be encoded in AI agents and how neuroscience can inspire more naturalistic AI [Hassabis et al., 2017].

**Central Pattern Generators**    In bio-inspired motor control, an extensive literature has built upon models of Central Pattern Generators (CPGs), which are neural circuits found across animal species that produce rhythmic activity in the absence of rhythmic inputs and sensory feedback. These circuits underlie fundamental rhythmic movements, including breathing, chewing, digesting, swimming, walking, and running. Computational models of CPGs have been successfully applied to diverse robotic bodies. For a review, refer to Ijspeert [2008] and Yu et al. [2014].

While our concept of a Neural Circuit Architectural Prior (NCAP) is related to CPGs, it differs in modeling constraints and formalization: (1) Neural-circuit-inspired connectivity can generate movement that includes, but is not limited to, rhythmic movement. In animals, of course, discrete movements (e.g. reflexes, reaching, sitting, jumping) are generated by neural circuits too. This work highlights the case study of swimming because *C. elegans* locomotion circuits are simple to explain, but related works have also explored circuits for reflexes [Liu et al., 2018], reaching [Schaal and Schweighofer, 2005], and decision making [Lechner et al., 2019, Hasani et al., 2020]. (2) CPG models vary significantly, with some based on neural networks (i.e. architectural priors) and many based on other formalisms including systems of coupled oscillators, vector maps, and finite-state machines [Ijspeert, 2008]. Neural-network-based CPGs further vary between the biophysically detailed, spiking, rate-coded, and population-level [Torres and Varona, 2012, Bing et al., 2018, Ijspeert, 2008]. This work adopts a discrete-time ANN formalism that is familiar to the broader AI community. In addition to pedagogical benefits, our formalism produces models that are fully differentiable, enabling us to tune parameters with the same reinforcement learning (RL) and evolution strategies (ES) algorithms commonly used in AI motor control. Further, our Swimmer NCAP architecture has the special property of being embeddable within a standard MLP of certain dimensions (Appendix B), enabling a direct, rigorous, and novel investigation of low-level connectivity in ANNs.

**Neuromechanical Models**    In computational neuroscience, models of neural circuits have been increasingly combined with models of biomechanics to study brain-body-environment interactions [Ausborn et al., 2021, Rybak et al., 2015, Danner et al., 2017]. Our work builds upon the rich insights gained from neuromechanical modeling, in particular of *C. elegans* swimming and steering circuits [Boyle et al., 2012, Demin and Vityaev, 2014, Izquierdo and Beer, 2015, Sarma et al., 2018]. However, while neuromechanical work primarily aims to elucidate biological principles and explain neural/behavioral data, our work has the distinct goal of translating insights to AI. To this end, we adopt the discrete-time ANN formalism as described above, and we also target a body from a standard AI benchmark: the DeepMind Control Suite [Tassa et al., 2020] built on the MuJoCo physics simulator [Todorov et al., 2012]. In contrast, the above-mentioned *C. elegans* works use biophysically realistic bodies and muscles. Our Swimmer body is significantly different from *C. elegans* in mechanics, degrees of freedom, and actuators; it is not obvious that *C. elegans* circuits would be useful, and our findings may encourage future work for other MuJoCo bodies as well. Ultimately, we hope our work can strengthen the bridge between the neuromechanical modeling and AI communities.

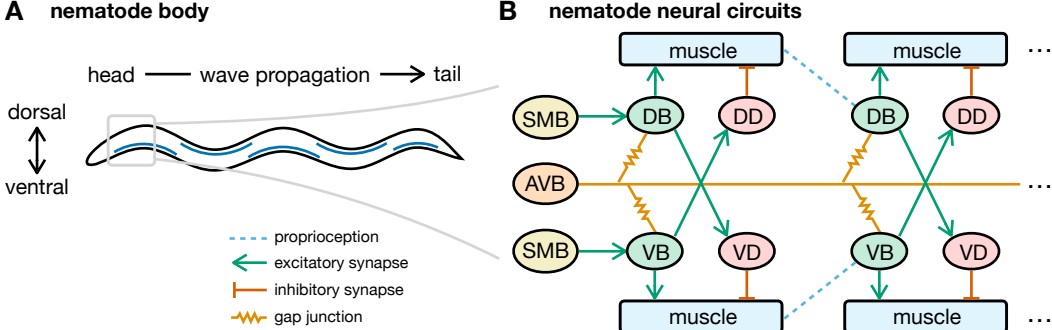

Figure 1: **Nematode. (A)** The nematode achieves forward locomotion through dorsal-ventral muscle contraction waves propagating down the body. **(B)** Muscle wave propagation, oscillation, steering, and speed control are coordinated by a highly stereotyped, modular, and repeated microcircuit. B neurons sense bending in the previous module and excite ipsilateral muscles, while inhibiting contralateral muscles via D neurons. Intrinsic oscillations in B neurons initiate waves. SMB neurons bias head/neck muscles for steering. AVB attenuates all B neurons via gap junctions for speed control.

## 3 Model

We translate nematode (*C. elegans*) locomotion circuits into an ANN model controlling a simulated Swimmer agent. In Section 3.1, we provide an overview of the nematode body and the neural circuitry underlying locomotion. In Section 3.2, we describe the integrator and oscillator units that serve as building blocks for our NCAP architecture. In Section 3.3, we formalize the observation and action space of the Swimmer agent, and we propose our NCAP architecture inspired by nematode circuits.

### 3.1 Nematode

The nematode *C. elegans* has served as a useful model organism within neuroscience because it is one of the simplest organisms with a nervous system. Moreover, it is unique in that its connectome, i.e. wiring diagram, has been completely mapped [Hall and Altun, 2008].

**Nematode Body** The nematode body is a 1 mm long, 50 μm diameter tapered cylinder (Figure 1A). It is made up of 959 somatic cells, of which 302 are neurons comprising the nervous system, of which 75 are motor neurons that innervate the 95 body wall muscles distributed along the body. Forward and backward thrust is produced via alternating dorsal-ventral muscle contraction waves propagating down the body in the direction opposite to the direction of motion. Steering is produced by differential activation of the 20 anterior muscles in the head and neck [Gjorgjieva et al., 2014].

**Nematode Neural Circuits** The nematode forward locomotion circuit is summarized below (Figure 1B). For an in-depth treatment, refer to Gjorgjieva et al. [2014] and Wen et al. [2018].

Muscle wave propagation is coordinated by 2 classes of neurons that innervate dorsal (D-) and ventral (V-) muscles. B neurons (DB and VB) act as both sensory and motor neurons, expressing stretch receptors in their dendrites to sense bending 200 μm anterior to their somas, and sending excitatory output (via ACh) to the muscles and to D neurons. D neurons (DD and VD) send inhibitory output (via GABA) to the muscles. This microcircuit is highly stereotyped, modular, and repeated down the length of the body, and its logic is interpretable. For a particular module, body bending in the previous module is sensed by B neurons, which then initiate bending on the same side (ipsilateral) while simultaneously inhibiting bending on the opposite side (contralateral) through D neurons.

Muscle wave initiation is generated by intrinsic oscillators. While proprioception-only circuits (with oscillators ablated) are capable of producing small waves on its own, oscillators are used initiate and entrain larger waves [Gjorgjieva et al., 2014]. These oscillators were long believed to only reside in the head and neck, but recently work has shown them to in fact also be present in the body as there is intrinsic oscillatory activity within the B neurons themselves [Wen et al., 2018].

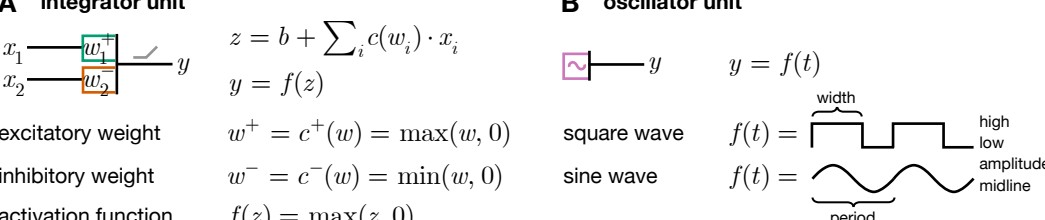

Figure 2: **Architectural Components. (A)** An integrator unit models a simple neuron. The graded input signals are multiplied by weights that represent synaptic efficacy and that are constrained to be either positive (excitatory, green boxes) or negative (inhibitory, red boxes). The graded output signal is produced by an activation function. **(B)** An oscillator unit produces driving signals much like intrinsic pacemaker cells and network-based oscillators. The graded output signal is generated by periodic functions, e.g. square waves and sine waves.

Steering is generated by the differential activation of SMB neurons biasing the head and neck muscles to bend dorsally or ventrally [Izquierdo and Beer, 2015].

Speed control is coordinated by the AVB command neuron, which is connected through gap junctions with all B neurons. When AVB is low, the resting membrane potentials of B neurons are hyperpolarized to prevent activation; when AVB is high, B neurons are free to activate.

## 3.2 Architectural Components

Our NCAP architecture is built from components that combine the discrete-time ANN formalism that is standard in machine learning with features from computational neuroscience like constraints on synapse sign (i.e. excitation vs. inhibition) and special cell types (i.e. intrinsic oscillator). The components are fully differentiable and therefore compatible with backpropagation-based learning algorithms, though not restricted to them.

**Integrator Units**    Signals in biological neural circuits are processed and integrated by neurons. The integrator unit[1] in Figure 2A is similar to the standard ANN model. The graded inputs $x_i$ are multiplied by synaptic weights $w_i$ to produce the membrane potential $z$, given the resting membrane potential $b$. A nonlinear activation function $f(z)$ produces the graded output $y$ based on the membrane potential. Unlike the standard ANN model, however, we constrain synaptic weights by a sign constraint function $c(w)$. This is done to reflect that in biological circuits a primary characteristic of a synapse is whether it is excitatory or inhibitory. In the standard model, synapses are initialized with random signs and are free to change during learning. We argue that constrained excitation/inhibition is fundamental for interpreting and modeling the logic of neural circuits, and we show in the ablation analysis that they are critical for learning in our architecture (Section 4.5).

**Oscillator Units**    Neural circuits often feature components with specialized dynamics, with oscillators being a prominent example [Grillner and El Manira, 2020]. An oscillator can be implemented through coupled activity between neurons or within a single neuron, similar to pacemaker cells in the heart [Bucher et al., 2015]. Oscillators serve as internal drivers of activity, exemplifying the fact (less appreciated within the ML community) that neural circuits are not exclusively driven by external inputs from the environment. The oscillator unit in Figure 2B uses a periodic function $f(t)$ to produce the graded output $y$. Example periodic functions include square wave and sine wave generators.

## 3.3 Swimmer

The Swimmer is an agent body in widely adopted AI motor control benchmarks, e.g. DeepMind Control Suite [Tassa et al., 2020] and OpenAI Gym [Brockman et al., 2016]. We target this standard body rather than a biorealistic body like previous work [Sarma et al., 2018, Izquierdo and Beer, 2015]

---

[1]Simple neurons are often approximated as a single integrator units [Torres and Varona, 2012]. However, sometimes neurons have multiple sites of integration, i.e. dendritic integration across multiple compartments. We prefer "integrator unit" to"neural unit" as a complex neuron may require multiple integrator units to model.

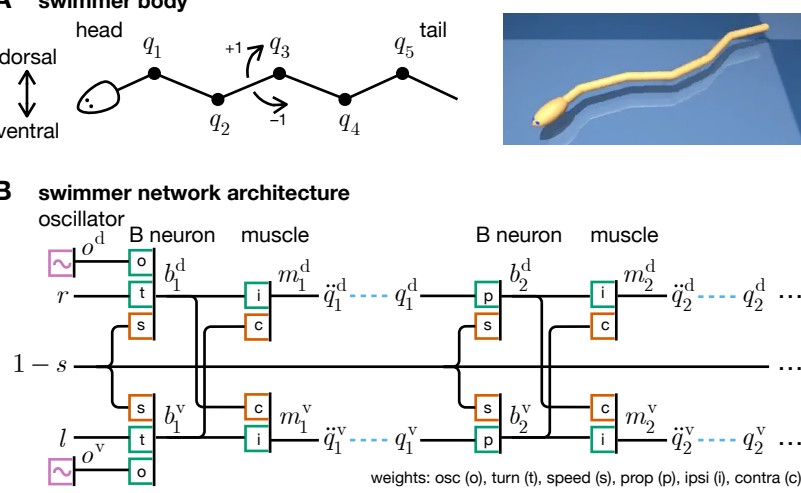

Figure 3: **Swimmer.** **(A)** The Swimmer has an articulated body with $N$ joints connecting $N+1$ links ($N = 5$ shown). Its observation space is normalized joint positions $\boldsymbol{q}$, and its action space is normalized joint accelerations $\ddot{\boldsymbol{q}}$. **(B)** Our network architecture closely conforms to the modular microcircuit of the nematode. Each module $i$ senses bending in the previous module $q_{i-1}$ and drives B neurons $b_i$ and muscles $m_i$, which are combined to create joint accelerations $\ddot{q}_i$.

in order to investigate the potential of biologically inspired network architecture in agents used by the broader AI community and to inspire future work in other MuJoCo bodies as well.

**Swimmer Body**    The Swimmer agent has an articulated body with $N$ joints connecting $N+1$ links (Figure 3A). Its movement is entirely within the $xy$-plane. Thrust is generated by the links pushing against the surrounding fluid, e.g. simulated via a high-Reynolds fluid drag model [Todorov et al., 2012]. The observation space consists of normalized joint positions $\boldsymbol{q} \in [-1, 1]^N$ between joint limits. The action space consists of normalized joint accelerations $\ddot{\boldsymbol{q}} \in [-1, 1]^N$ between maximum acceleration counterclockwise and clockwise, respectively.

**Swimmer Network Architecture**    Our NCAP architecture is best explained visually (Figure 3B).

For muscle wave propagation, signals are integrated in B neurons and muscles; D neurons mainly serve to convert opposite-side B neuron signals from excitatory to inhibitory, and their role can be replicated directly in the muscle integrator units. We model $N$ modules to control each of the $N$ joints. For a particular module $1 \le i \le N$, the previous module joint position $q_{i-1}$ is split into dorsal $q_{i-1}^{\mathrm{d}} \in [0, 1]$ and ventral $q_{i-1}^{\mathrm{v}} \in [0, 1]$ components, in order to mirror signals from proprioceptive stretch receptors that are sensitive to bending on one side. B neurons are modeled as integrator units with outputs $b_i^{\mathrm{d}}$ and $b_i^{\mathrm{v}}$, which receive same-side excitatory proprioceptive inputs. Muscles are modeled as integrator units with outputs $m_i^{\mathrm{d}}$ and $m_i^{\mathrm{v}}$, which receive same-side (ipsilateral) excitatory B neuron input as well as opposite-side (contralateral) inhibitory B neuron input. Finally, the joint acceleration $\ddot{q}_i$ is calculated from dorsal and ventral muscle outputs, which act antagonistically.

For muscle wave initiation, the first module B neurons $b_1^{\mathrm{d}}$ and $b_1^{\mathrm{v}}$ receive inputs from oscillators $o^{\mathrm{d}}$ and $o^{\mathrm{v}}$, respectively, instead of proprioception. We use square wave generators acting in anti-phase.

For steering, SMB outputs are modeled as a right turn signal $r \in [0, 1]$ and a left turn signal $l \in [0, 1]$, which serve as additional excitatory inputs to first module B neurons $b_1^{\mathrm{d}}$ and $b_1^{\mathrm{v}}$, respectively.

For speed control, AVB outputs are modeled as a speed signal $s \in [0, 1]$. To approximate the effect of gap junctions, such that $s = 0$ represents stopping and $s = 1$ represents maximum speed, $1 - s$ serves as an additional inhibitory input to all B neurons.

The complete architecture for module $i$ is therefore:

$$q_{i-1}^{\text{d}} = \max(q_{i-1}, 0) \qquad\qquad q_{i-1}^{\text{v}} = \max(-q_{i-1}, 0)$$

$$b_i^{\text{d}} = f\left(w_{\text{prop}}^+ q_{i-1}^{\text{d}} + w_{\text{speed}}^-(1-s)\right) \qquad b_i^{\text{v}} = f\left(w_{\text{prop}}^+ q_{i-1}^{\text{v}} + w_{\text{speed}}^-(1-s)\right)$$

$$m_i^{\text{d}} = f\left(w_{\text{ipsi}}^+ b_i^{\text{d}} + w_{\text{contra}}^- b_i^{\text{v}}\right) \qquad m_i^{\text{v}} = f\left(w_{\text{ipsi}}^+ b_i^{\text{v}} + w_{\text{contra}}^- b_i^{\text{d}}\right)$$

$$\ddot{q}_i = \min(m_i^{\text{d}}, 1) - \min(m_i^{\text{v}}, 1)$$

with special B neuron integrator units for $i = 1$:

$$b_1^{\text{d}} = f\left(w_{\text{osc}}^+ o^{\text{d}} + w_{\text{turn}}^+ r + w_{\text{speed}}^-(1-s)\right) \qquad b_1^{\text{v}} = f\left(w_{\text{osc}}^+ o^{\text{v}} + w_{\text{turn}}^+ l + w_{\text{speed}}^-(1-s)\right)$$

We use weight sharing such that weights with the same name are shared across modules as well as within each module across sides. We initialize all weights with the correct signs and magnitudes of 1.

## 4 Experiments

**Learning Formalization**  We consider an agent formalized as a policy function $\pi_{\boldsymbol{\theta}}(\boldsymbol{a}_t|\boldsymbol{s}_t)$ that maps states $\boldsymbol{s}_t$ to actions $\boldsymbol{a}_t$, and which is represented by an ANN parameterized by weights $\boldsymbol{\theta}$ with our architecture described in Section 3.3. We consider the standard agent-environment interaction model formalized as a Markov Decision Process (MDP). At every timestep $t$, the agent in state $\boldsymbol{s}_t$ takes an action according to its policy $\boldsymbol{a}_t \sim \pi_{\boldsymbol{\theta}}(\boldsymbol{a}_t|\boldsymbol{s}_t)$, receives a reward $r_t$, and transitions to a new state $\boldsymbol{s}_{t+1}$. Policy parameters $\boldsymbol{\theta}$ are optimized to maximize the discounted return $\sum_{t=0}^{T} \gamma^t r_t$, where $T$ is the horizon of the episode, and $0 < \gamma \leq 1$ is the discount factor.

**Learning Setup**  We implement the Swimmer using the standard $N = 5$ body in the DeepMind Control Suite [Tassa et al., 2020] built upon the MuJoCo physics simulator [Todorov et al., 2012]. We train the agent to swim using shaped rewards proportional to swimming speed (Appendix A.1).

**Learning Algorithms**  We compare backpropagation-based RL algorithms (PPO, DDPG) and a derivative-free ES algorithm (OpenAI-ES) for tuning parameters in the architecture (Appendix A.2).

### 4.1 Performance and Data Efficiency

*How well and data efficiently does learning occur in NCAP vs. MLP architectures?*

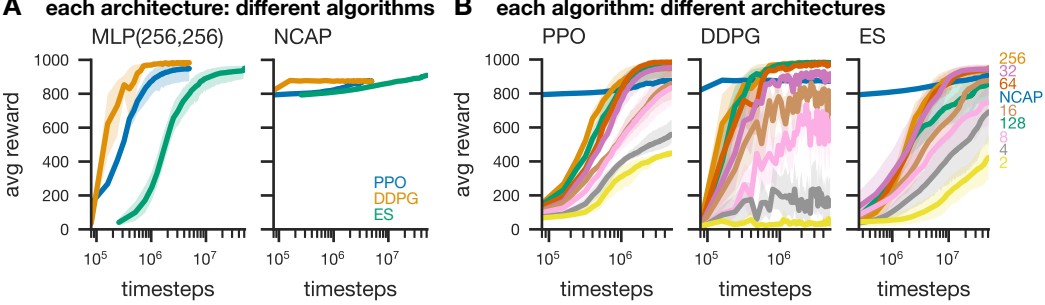

Figure 4: **Performance and Data Efficiency. (A)** Comparison across architectures, using different algorithms. Our architecture starts with high reward and improves with learning, achieving significantly better data efficiency and comparable performance. **(B)** Comparison across algorithms, using different architectures. Our architecture with 4 parameters overperforms small MLPs (e.g. MLP(2,2) has 74 parameters) and is comparable to large MLPs (e.g. MLP(256, 256) has 73,226 parameters). Therefore, structured connectivity (not simply parameter count) matters for performance. Plots show averages over 10 random seeds (solid lines) and 95% bootstrap confidence intervals (shaded areas).

First, we compare different learning algorithms for either MLP or NCAP architectures (Figure 4A). We use an MLP with 2 hidden layers of dimensions (256, 256) and ReLU nonlinearities. We find that our NCAP architecture achieves substantially higher initial performance than MLPs as well as comparable asymptotic performance with MLPs, demonstrating the effectiveness of prior knowledge encoded in network architecture. Our NCAP architecture shows reduced variance during learning between trials with different random seeds, as well as reduced differences in asymptotic performance between algorithms. For both MLP and NCAP, ES requires roughly an order of magnitude more data to achieve comparable performance with the RL algorithms, consistent with previous work [Salimans et al., 2017]. Qualitatively, both MLP and NCAP architectures yield reasonable swimming movement (Videos 1A-B), though NCAP produces waves with large amplitudes resembling *C. elegans* movement, while MLP produces waves with small amplitudes more resembling tadpoles. This different movement shape explains the slightly lower asymptotic performance for NCAP because the body's direction of travel is less correlated with the head orientation, which is relevant for how rewards are calculated (Appendix A.1, Videos 1C). We note that we simplified our design from actual *C. elegans* circuits for pedagogical reasons (Section 4.5), and our goal is not to solve this swimming task *per se* but rather to investigate the advantages of biologically inspired network architecture more generally. *C. elegans* circuits are not optimized for fast swimming with few segments (Section 4.4); future work may propose architectures better for this specific task, e.g. using larval zebrafish circuits.

Second, we compare different architectures for each learning algorithm (Figure 4B). We use MLPs with 2 hidden layers of varying dimension sizes and ReLU nonlinearities. We find that performance deteriorates across all algorithms for MLPs as hidden dimensions become smaller, with some algorithms like DDPG deteriorating dramatically. However, our NCAP architecture with 4 parameters overperforms small MLPs and is comparable to large MLPs. This is especially notable as the smallest MLP(2,2) has 74 parameters (1 order of magnitude more than NCAP) and the largest MLP(256,256) has 73,226 parameters (2 orders of magnitude more than NCAP). This suggests that the relatively simple structure of our NCAP architecture provides highly effective inductive biases.

## 4.2 Parameter Efficiency

*How valuable are parameters in NCAP vs. MLP architectures?*

We compare the asymptotic performance divided by parameter count for each architecture (Figure 5). Across all algorithms, our NCAP architecture achieves more than an order of magnitude better parameter efficiency than MLPs of any size. A parameter in our NCAP architecture is "worth more" than one in MLPs.

## 4.3 Interpretability

*How interpretable are unit dynamics in our NCAP architecture?*

We visualize network unit dynamics during a task (Videos 2). Since our NCAP architecture is a sparsely connected, modular network where units play constrained roles (e.g. excitatory or inhibitory), it is easier to relate unit dynamics to agent behavior, which facilitates debugging and engineering constraints for safety.

## 4.4 Transfer

*How well does our NCAP architecture adapt to new bodies?*

We leverage the modular structure of our NCAP architecture to adapt a trained network for $N = 5$ joints to bodies with different $N'$ joints by adding/removing modules (Figure 6). Even without further training, we achieve robust swimming for a wide range of $N'$ (Videos 3). Interestingly, performance is *better* for longer bodies, possibly reflecting that nematode circuits are evolved for a more segmented body. This kind of zero-shot transfer is not possible with monolithic, densely connected MLP architectures.

**parameter count & efficiency**

| model | params | $\log_2$ ( reward / params ) |
|---|---|---|
| (256,256) | 73,226 | -6.3 ± 0.2 |
| (128,128) | 20,234 | -4.4 ± 0.2 |
| (64,64) | 6,026 | -2.6 ± 0.1 |
| (32,32) | 1,994 | -1.1 ± 0.2 |
| (16,16) | 746 | 0.2 ± 0.3 |
| (8,8) | 314 | 1.4 ± 0.4 |
| (4,4) | 146 | 1.9 ± 0.3 |
| (2,2) | 74 | 2.6 ± 0.2 |
| **NCAP** | **4** | **7.8 ± 0.1** |

Figure 5: **Parameter Efficiency.** Asymptotic reward per unit parameter. Our model achieves better parameter efficiency than MLPs of all tested sizes.

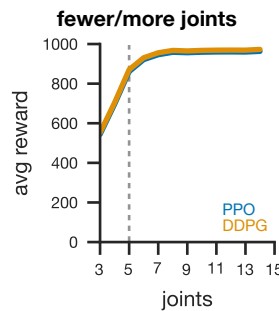

Figure 6: **Transfer.** After training with $N = 5$ joints, our model can transfer to different $N'$ by adding/removing modules.

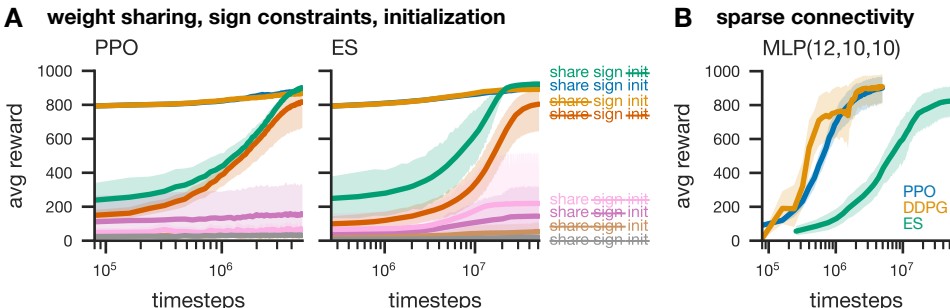

Figure 7: **Ablations. (A)** Ablations of weight sharing, sign constraints, and initialization in different combinations. Sign constraints are crucial for learning. Weight initialization contributes to good initial performance. Weight sharing yields a small gain. **(B)** Ablation of sparse connectivity yields an equivalently sized MLP (Appendix B). Learning is restored without sparse connectivity.

## 4.5 Ablations

*What are the effects of various features of our NCAP architecture on performance and learning?*

First, we investigate the role of weight sharing, sign constraints, and weight initialization (Figure 7A). Without weight sharing, weights across modules and across sides are separate parameters, increasing the total number of parameters from 4 to 30. Without sign constraints, the identity function is applied to weights instead of the constraint function. Without typical weight initialization, weight magnitudes are initialized through a uniform random distribution within $[0, 1]$, rather than at 1. If using sign constraints, weights are always initialized with the appropriate sign; otherwise, signs are chosen randomly with equal probability. We find that sign constraints are crucial for learning. Without appropriate sign constraints, our NCAP architecture fails to learn during the allotted timesteps, for both RL and ES algorithms. With appropriate sign constraints, even if the weights magnitudes are not initialized large, our NCAP architecture will learn. Weight initialization is responsible for good initial performance. Weight sharing has a smaller, but identifiable, contribution to data efficiency.

Second, we investigate the role of sparse connectivity that arises from the natural structure of neural circuits (Figure 7B). Our Swimmer architecture has the special property that it can be completely embedded within an MLP with 3 hidden layers of dimensions (12, 10, 10) and ReLU nonlinearities (Appendix B). Specifically, after ablating sign constraints and weight sharing, our architecture is identical to this MLP with highly pruned connectivity (mostly weights of 0). We remove this sparsity and find that the MLP can learn the task with similar asymptotic performance as our NCAP.

Taken together, our results suggest that constrained excitation/inhibition is an important consideration in small, sparse architectures like our NCAP, but less important in MLPs. This may be related to the "Lottery Ticket Hypothesis" [Frankle and Carbin, 2019], which suggests that, upon initialization, the MLPs already contain subnetworks with initial weights *and signs* that do most of the work for learning, i.e. they are "winning tickets"; imposing sparsity eliminates these overlapping subnetworks.

## 5 Discussion

We asked what advantages biologically inspired ANN architecture can provide in the domain of motor control. Through our case study translating *C. elegans* locomotion circuits into an ANN model, we found that biologically inspired ANN architecture can achieve comparable asymptotic performance to MLPs with significantly improved initial performance, data efficiency, parameter efficiency, interpretability, and transfer. Therefore, while *tabula rasa* architectures are general, Neural Circuit Architectural Priors (NCAP) can provide useful inductive biases for motor control.

Nevertheless, our case study here was limited: it focused on a relatively simple body and circuit, and it prioritized pedagogical simplicity over performance optimization and behavioral complexity. Future work should explore additional bodies, movement types, and circuit modalities:

(1) *Bodies*. Many animals generate movement through highly structured pattern generator circuits, including quadrupeds (e.g. mice, cats, dogs, horses) and bipeds (e.g. birds, humans) [Ijspeert, 2008].

In particular, quadrupeds are commonly studied in both neuroscience and robotics [Rybak et al., 2015, Danner et al., 2017, Iscen et al., 2019]. Quadruped locomotion circuits are located primarily within the spinal cord and produce robust gaits (e.g. walk, bound, gallop) even when top-down connections from the brain are lesioned [Buschmann et al., 2015]. Insights from these circuits could be translated to AI motor control bodies, and a Quadruped NCAP architecture may use similar integrators, oscillators, and constrained connections as our Swimmer NCAP.

(2) *Movement Types*. Neural circuits coordinate diverse animal movements, both rhythmic (e.g. breathing, chewing, swimming, walking) and discrete (e.g. reflexes, reaching, sitting, jumping). Therefore, NCAP architectures that closely resemble neural circuit mechanisms could in principle be applied to a variety of additional movement types.

(3) *Circuit Modalities*. Biologically inspired ANN architecture may also provide useful inductive biases for upstream tasks involving perception and decision making. In animals, lower-level pattern generator circuits are modulated by higher-level control circuits. For instance, locomotion speed and direction are modulated by orientation and escape circuits in response to visual, auditory, and tactile stimuli. Such higher-level circuits could inspire NCAP architectures for additional modalities.

Ultimately, our work suggests a way of advancing AI and robotics research inspired by systems neuroscience and encourages future work in more complex embodied control.

## Acknowledgments and Disclosure of Funding

Thanks to Sergey Shuvaev, Pavel Tolmachev, Liam McCarty, Polina Kirichenko, Pavel Izmailov, Daniel Barabasi, Christopher Langdon, and Josh Merel for insightful discussions and manuscript feedback. This work was supported by Schmidt Futures (N.X.B.); the G. Harold and Leila Y. Mathers Charitable Foundation (A.M.Z.); the Eleanor Schwartz Charitable Foundation (A.M.Z.); NIH grants S10OD028632-01 and RF1DA055666 (T.A.E.); and the Alfred P. Sloan Foundation Research Fellowship (T.A.E.).

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

# A Experimental Details

## A.1 Tasks

The `swim` task aims to test the agent's ability to swim forwards at a desired speed. It returns a smooth reward that is 0 when stopped or moving backwards, and rises linearly to and saturates at 1 when swimming at the desired speed.

## A.2 Learning Algorithms

**Proximal Policy Optimization (PPO)**   [Schulman et al., 2017] A model-free, on-policy, policy gradient RL method. It uses a clipped surrogate objective to limit the size of policy change at each step, thereby improving stability. Since it assumes stochastic policies, we perturb the deterministic actions with Gaussian noise $\epsilon_i \sim \mathcal{N}(0, \sigma^2)$.

**Deep Deterministic Policy Gradient (DDPG)**   [Lillicrap et al., 2019] A model-free, off-policy, policy gradient RL method. It uses off-policy data and the Bellman equation to learn the Q-function, which is iteratively used to improve the policy.

**Evolution Strategies (ES)**   [Salimans et al., 2017] An evolutionary black-box optimization method. It creates a population of policy parameter variants through perturbations with Gaussian noise, then combines them through averaging, weighted by the return collected across episodes.

## A.3 Implementation

**Libraries**   Neural networks were implemented in PyTorch (BSD license) [Paszke et al., 2019]. The RL algorithms were implemented using Tonic (MIT license) [Pardo, 2021]. The ES algorithm was implemented using ES Torch (MIT license) [Karakasli, 2020].

**Computational Resources**   Training was performed on a high performance computing cluster running the Linux Ubuntu operating system. RL algorithm training runs were parallelized over 8 cores, while ES algorithm runs were parallelized over 32 cores.

## A.4 Hyperparameters

**RL Algorithms**   Standard hyperparameters for PPO and DDPG in Tonic [Pardo, 2021] at commit `48a7b72`; timesteps, 5e6.

**ES Algorithm**   Population size, 256; noise standard deviation $\sigma$, 0.02; L2 weight decay, 0.005; optimized, Adam; learning rate, 0.01; timesteps, 5e7.

**NCAP Swimmer**   Oscillator, square wave, period 60 timesteps, width 30 timesteps.

# B Swimmer Architecture Details

Our NCAP architecture has the special property that it can be completely embedded within a fully connected MLP of 3 hidden layers and ReLU nonlinearities. This enables us to "interpolate" between our NCAP architecture and the MLP architecture, conducting a rigorous analysis of how various features of our architectural prior contribute to performance and learning.

By rearranging terms in the Swimmer network architecture diagram (Figure 3B), we arrive at the following network ($N = 5$ shown):

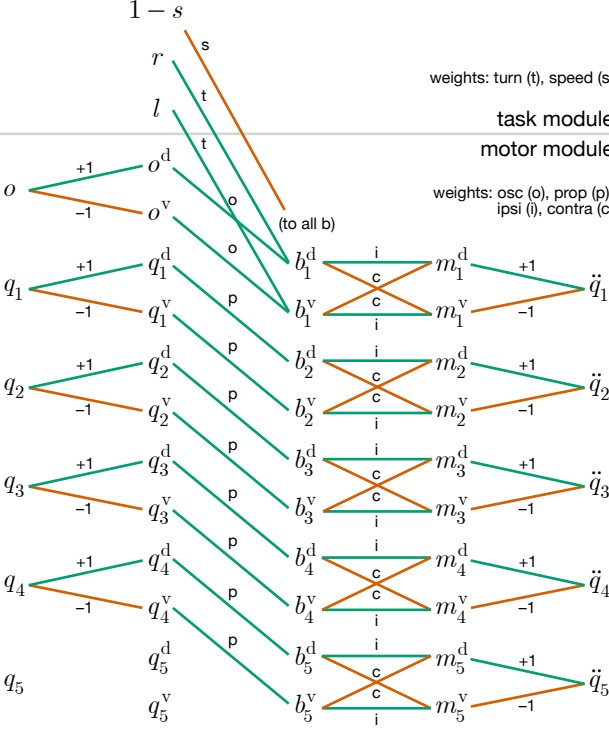

By removing weight sharing, sign constraints, and sparse connectivity, we arrive at a fully connected MLP of 3 hidden layers ($N = 2$ shown):

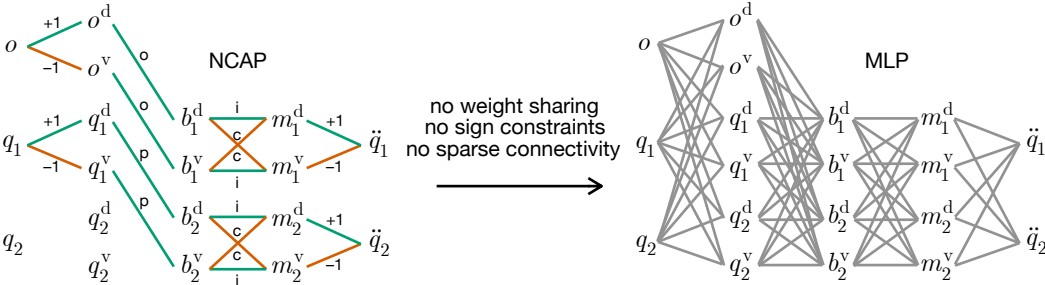

For $N = 5$, the resulting MLP has hidden layers of dimensions (12, 10, 10).

