# OpenReview forum: "Neural Circuit Architectural Priors for Embodied Control"
_NeurIPS.cc/2022/Conference — NeurIPS 2022 Accept_

### Official Review · Reviewer_uQAH · 2022-07-02

**Rating:** 6
**Confidence:** 3
**Soundness:** 3 good
**Presentation:** 3 good
**Contribution:** 2 fair

**Summary:**


Neural Circuit Architectural Priors for Embodied Control
In this paper, the authors are inspired by the Caenorhabditis elegans locomotion circuits, and design a structurally similar neural network structure. This prior structure provides valuable starting motion control knowledge and enables faster training in the task of the swimmer from the deepmind control suite.


**Questions:**

1) From a neuroscience’s perspective, maybe the authors can comment on the connection between the proposed algorithm and some other neuro-inspired network structures? For example the feedback alignment network in [3], [4] and [5]
2) Back in the “Strengths And Weaknesses”, I mentioned the lack of experiments. Could the authors provide some comments on whether or not the algorithm can be applied to other creatures and other tasks?
3) The periodic phase generation module, while not exactly the same, was proposed and used in some character animation research such as [1], [2]. These are some literature I think is related to the oscillator units discussed in the paper.


**Limitations:**


I don’t see any potential negative societal impact in this work.

[1] Van de Panne, Michiel, Ryan Kim, and Eugene Fiume. "Virtual wind-up toys for animation." In Graphics Interface, pp. 208-208. CANADIAN INFORMATION PROCESSING SOCIETY, 1994.
[2] Holden, Daniel, Taku Komura, and Jun Saito. "Phase-functioned neural networks for character control." ACM Transactions on Graphics (TOG) 36, no. 4 (2017): 1-13.
[3] Lillicrap, Timothy P., Daniel Cownden, Douglas B. Tweed, and Colin J. Akerman. "Random synaptic feedback weights support error backpropagation for deep learning." Nature communications 7, no. 1 (2016): 1-10.
[4] Nøkland, Arild. "Direct feedback alignment provides learning in deep neural networks." Advances in neural information processing systems 29 (2016).
[5] Frenkel, Charlotte, Martin Lefebvre, and David Bol. "Learning without feedback: Fixed random learning signals allow for feedforward training of deep neural networks." Frontiers in neuroscience 15 (2021): 629892.



**Strengths And Weaknesses:**


Originality: This paper draws a very interesting connection between neuroscience and reinforcement learning. Since I have no previous knowledge in biology or neuroscience, I cannot comment on how novel this idea is. From the reinforcement learning point of view, I think it’s pretty novel and inspiring.

Quality and clarity: The paper is well written. The related work is pretty adequate, and the figures in the paper (for example Figure 1 and Figure 2) are pretty intuitive and easy to understand.
The experiments are well designed, with every experiment having clear purposes and explanations.

Significance: How well does it scale to high-dimensional agents? How well does it scale to other tasks? I think these are some of the questions that would greatly affect how significant the proposed algorithm is in the community of reinforcement learning (I cannot comment on the neuroscience part unfortunately).
Without experiments on more creatures (for example the ant like creatures, the fish like creatures in the deepmind control suite), or more experiments for different tasks that aim to show the locomotion skills not only on swimming but also on reaching, avoiding, and even 3D swimming, the significance of the algorithm remains unclear.

---

> ### Author Response · Authors · 2022-08-02
> **Response to R4**
>
> Thank you for your feedback and support! It was valuable to hear your comments from the non-neuro, reinforcement learning point of view, and we are quite pleased that you found the work "novel and inspiring." Our goal in this paper is exactly to build such bridges between neuro and AI, which is one of the reasons we highlight the C. elegans case study and provide extensive neuro background. We hope these responses satisfactorily answer your questions:
>
> **Context within neuro-AI (#1): "Maybe the authors can comment on the connection between the proposed algorithm and some other neuro-inspired network structures?"**
>
> Sure, the works you referred to in [3, 4, 5] aim to approximate backpropagation in layered networks using biologically plausible learning rules, since the mechanisms needed to compute exact backpropagation in ANNs are considered implausible within the brain. Our Swimmer NCAP architecture doesn't share the same aims, and we trained using both (exact) backprop-based RL algorithms and a backprop-free ES algorithm.
>
> **Scaling to high-dimensional agents (#2): "Could the authors provide some comments on whether or not the algorithm can be applied to other creatures and other tasks?"**
>
> Yes, we believe that biologically inspired network architecture will be useful for other bodies and movements too! Please kindly refer to our response to **R1** about how a Quadruped NCAP could take a similar approach as our Swimmer NCAP.
>
> **Oscillator units (#3): "[1, 2] are some literature I think is related to the oscillator units discussed in the paper"**
>
> Thanks for pointing to these works. Yes, there have been many works that use oscillators to drive motor control dynamics, especially in classical robotics and trajectory-based priors. In a way, our model can be seen as a hybrid approach that combines neural networks with specialized components like oscillators, but we certainly can't claim novelty to oscillators! However, the core contribution of this work, beyond any specific component, lies in demonstrating how neural circuits can inspire AI motor control architectures.

---

### Official Review · Reviewer_8ee3 · 2022-07-08

**Rating:** 4
**Confidence:** 3
**Soundness:** 2 fair
**Presentation:** 2 fair
**Contribution:** 2 fair

**Summary:**

This paper introduces a new neural network architecture, "NCAP" (Neural Circuit Architectural Priors), for performing a locomotion task (the Swimmer task from the DeepMind Control Suite).  The network is inspired by the locomotion circuits of C. elegans.  The authors train both NCAP, as well as various MLP architectures to perform the Swimmer task, and do so using 3 different RL training methods (PPO, DDPG and ES, an evolutionary black-box optimization method).  The NCAP architecture consistently achieves higher average reward earlier in training compared to the MLP architectures, while having significantly fewer trainable parameters.

**Questions:**

My questions were outlined above, but I repeat these here for completeness:

1. Clarity: Please clarify what the state at time $t$, $s_{t}$, corresponds to on line 240.  You give the inputs to NCAP in the equations of Section 3, but please clarify the exact inputs you are giving to the MLP comparison networks.  Is $s_{t}$ for the MLP networks just $\{o, q_{1}, q_{2} ,..., q_{N}\}$ or does this also include the speed $s$ and $r$ and $l$?  From Appendix B, it looks like you don't give the speed $s$ and $r$ and $l$ to the MLP.
2. Clarity/quality: It seems strange to me that speed is given as an input to NCAP, given that reward is based on speed.  This is especially true if you only give speed as an input to NCAP and not to the MLP networks.  To what extent can the early success of NCAP relative to the MLP networks be attributed to the fact that speed is given as an input to NCAP?  It would have been nice to see an ablation study where the authors removed the speed input in order to get a sense of how crucial this input is.  Furthermore, it would nice to have been able to see an expanded x-axis for Figure 4A so as to know when NCAP performance first reached the avg reward of 800 (was NCAP achieving this reward from time=0 or time=$10^{4}$?)
3. Significance: while the results of NCAP on the Swimmer task are impressive, I am not sure of the broader significance of this paper.  This is especially true because of the fact that it appears that the asymptotic performance of NCAP is lower than that of some MLP architectures (Figure 4).  Are there other tasks where the NCAP architecture can be applied?  Or is the take-away message from this paper that sign constraints on weights in neural networks are important, and enable faster learning?  I am excited to hear what the authors have to say on this point: what do you think the broader implications of your paper are - beyond the Swimmer task?

**Limitations:**

The Discussion section for this paper was a little lackluster, and the authors could have been more explicit about the limitations of their work. For example, one limitation is that you've only demonstrated the utility of the NCAP architecture on the Swimmer task.  Another limitation is that the asymptotic performance of NCAP seems to be lower than that of the MLP architectures (Figure 4).

**Strengths And Weaknesses:**

Strengths:
1. Originality: despite the connectome of C. elegans having been available for years, I don't think there are many papers trying to analyze the key components of the anatomy of C. elegans that enable these worms to learn efficiently, as well as the implications of the C. elegans connectome for building AI systems.  In this sense, the paper tackles interesting, under-explored questions
2.  Quality: the paper is relatively well written.  The authors are thorough in their review of Related Work, and the figures - especially the early figures that give the overview of their framework and the nematode - are clear and easy to understand.

Weaknesses:
1. Clarity: Please clarify what the state at time $t$, $s_{t}$, corresponds to on line 240.  You give the inputs to NCAP in the equations of Section 3, but please clarify the exact inputs you are giving to the MLP comparison networks.  Is $s_{t}$ for the MLP networks just $\{o, q_{1}, q_{2} ,..., q_{N}\}$ or does this also include the speed $s$ and $r$ and $l$?  From Appendix B, it looks like you don't give the speed $s$ and $r$ and $l$ to the MLP.
2. Clarity/quality: It seems strange to me that speed is given as an input to NCAP, given that reward is based on speed.  This is especially true if you only give speed as an input to NCAP and not to the MLP networks.  To what extent can the early success of NCAP relative to the MLP networks be attributed to the fact that speed is given as an input to NCAP?  It would have been nice to see an ablation study where the authors removed the speed input in order to get a sense of how crucial this input is.  Furthermore, it would nice to have been able to see an expanded x-axis for Figure 4A so as to know when NCAP performance first reached the avg reward of 800 (was NCAP achieving this reward from time=0 or time=$10^{4}$?)
3. Significance: while the results of NCAP on the Swimmer task are impressive, I am not sure of the broader significance of this paper.  This is especially true because of the fact that it appears that the asymptotic performance of NCAP is lower than that of some MLP architectures (Figure 4).  Are there other tasks where the NCAP architecture can be applied?  Or is the take-away message from this paper that sign constraints on weights in neural networks are important, and enable faster learning?  I am excited to hear what the authors have to say on this point: what do you think the broader implications of your paper are - beyond the Swimmer task?

Minor points:
1.  Clarity: Are there not 6 parameters for NCAP?  $w_{prop}, w_{speed}, w_{turn}, w_{osc}, w_{ipsi}, w_{contra}$?  In Table 5, you say that there are 4 parameters
4. Clarity: there are many constrained parameters and outputs for the NCAP network.  While you clarify how weights are appropriately constrained to have the desired signs, I wasn't clear on how you constrain the output of the network to be in [-1, 1] (line 214).

Extremely minor point:
1. Quality (I only include this here because I noticed it when I zoomed into the figure and I would want to correct it if this was my paper): the x tick labels on Figure 6 are not vertically aligned

---

> ### Author Response · Authors · 2022-08-02
> **Response to R3**
>
> Thank you for your insights and depth of feedback! We appreciated your comments about the paper's quality and originality. We hope these responses satisfactorily address the raised questions, especially those about broader implications.
>
> **Broader implications (#3): "What do you think the broader implications of your paper are, beyond the Swimmer task?"**
>
> The goal of this paper is to start a conversation about how the fields of neuromechanical modeling and AI motor control can inform each other. As you point out, the connectome of C. elegans has been available for years, but interesting questions related to how understanding neural circuits can inform AI systems have remained under-explored. The broader implications of this paper are fourfold:
>
> 1) By demonstrating the advantages that biologically inspired network architecture can provide relative to MLPs (comparable asymptotic performance with significantly improved initial performance, data efficiency, parameter efficiency, interpretability, and transfer), we motivate future work exploring neural-circuit-inspired architectural priors in other AI motor control bodies. As **R1** notes, adding prior structure into ANNs is an important problem for continuous control / robotics, and our work provides a novel perspective on how this can be done. Please kindly refer to our response to **R1** about how a Quadruped NCAP could take a similar approach as our Swimmer NCAP. Indeed, we are pursuing such directions ourselves, but tackling additional bodies is necessarily outside the scope of this current paper.
>
> 2) By adopting a formalism that combines the standard discrete-time ML framework with features from computational neuroscience, we demonstrate the value of biologically inspired synapse sign constraints (i.e. excitation vs. inhibition) and special cell types (i.e. intrinsic oscillators). It is not common within ML to incorporate such features within ANNs, but our results encourage further exploration.
>
> 3) By systematically investigating the effects of MLP architecture size (Fig 4B) and MLP-NCAP interpolation (Appendix B), we raise interesting questions about MLP training dynamics. In particular, we authors have had spirited discussions about why small MLPs don't learn (Fig 4B) and whether large MLPs learn small-network solutions embedded within them (Appendix B). We don't have the answers for these questions.
>
> 4) By walking readers through a detailed construction process from neural circuits to ANN architecture, we hope this paper to be a good pedagogical resource demonstrating how systems neuroscience can guide neural network modeling. We believe that C. elegans was an ideal case study to start with, due to the availability of its connectome, the simplicity of its locomotion circuits, and the similarity of its body to simple AI motor control benchmark bodies. Already **R4**, with a primary background in RL, has found this work "novel and inspiring"; other AI/ML researchers at NeurIPS may also find this work's perspective to be original and surprising.
>
> **Discussion section: "The authors could have been more explicit about the limitations of their work"**
>
> Thank you for your suggestion, we agree. We already discussed the asymptotic performance results a bit  (lines 265-271), but we have improved the discussion section to mention limitations more explicitly.
>
> **Parameter count: "Are there not 6 parameters for NCAP?"**
>
> Yes, there are 6 parameters total in our model: 4 motor module (prop, osc, ipsi, contra) and 2 task module (speed, turn) — see Appendix B. In Table 5, we say 4 parameters because we are training only the motor module for the swim task (no turning or speed control necessary).
>
> **Observation space (#1, #2): "What the state at time t corresponds to?" / "Speed is given as an input to NCAP?"**
>
> Yes, you are correct that input to the MLPs is o, q1, q2, …, qN. The NCAP model receives the exact same input. In particular, speed (s) and turn (r, l) control signals are not used when training only the motor module; they are currently only used in the interpretability section, which demonstrates a navigate task.
>
> **Action space: "I wasn't clear on how you constrain the output of the network to be in [-1, 1] (line 214)"**
>
> We compute the joint accelerations by taking the difference between the muscle activations on each side of the body, which we clamped to be each within [0, 1]. We've updated the manuscript to clarify this (line 235).
>
> **Figure labels: "The x tick labels on Figure 6 are not vertically aligned"**
>
> Weird, something happened in the pdf conversion step. Thanks for pointing it out, we'll fix it in the final manuscript!

---

> > ### Comment · Reviewer_8ee3 · 2022-08-09
> > **Response to authors**
> >
> > I thank the authors for their response to my review.  I am grateful to them for describing what they think the contributions of their paper are beyond the Swimmer task, however, if their goal is to `walk readers through a detailed construction process from neural circuits to ANN architecture`, I think I am inclined to agree with R2's takeaway message when they say that this work could have a greater impact if refactored as a contribution to neuroscience rather than AI.  Alternatively, the work would be enhanced if the authors did more a systematic investigation to understand `why small MLPs don't learn (Fig 4B) and whether large MLPs learn small-network solutions embedded within them (Appendix B). We don't have the answers for these questions.` I also had hoped that the authors would answer my comment `Furthermore, it would nice to have been able to see an expanded x-axis for Figure 4A so as to know when NCAP performance first reached the avg reward of 800 (was NCAP achieving this reward from time=0 or time=?)`.  As such, I will leave my score as is.

---

### Official Review · Reviewer_QpQM · 2022-07-11

**Rating:** 4
**Confidence:** 4
**Soundness:** 2 fair
**Presentation:** 3 good
**Contribution:** 1 poor

**Summary:**

The authors introduce NCAP, a neural network architecture inspired by computational models of actual nematode locomotion. Experiments are done on the "swimmer" environment, which bears some relationship to nematode locomotion. They show that when optimized for forward velocity (via RL or ES), NCAP has much better initial performance than conventional architectures, despite having far fewer numerical parameters. Ablations reveal than the sign constraints in particular are crucial for performance. Finally, the authors show that training transfers to bodies with a different number of joints.

**Questions:**

1) Would NCAP work when optimizing a different reward function, like reaching a goal location?

2) Would NCAP work for non-swimmer environments? Particularly those with similarly structured action spaces (joint accelerations).

3) Why should we care about the parameter counts of small models independent of data efficiency? Even the largest model considered here isn't bottlenecked by memory or compute, and it's unclear these gains would hold for larger models. In particular, why is the reward/params metric in Figure 5 meaningful?

4) What do the transfer results (Figure 6) look like for a fresh initialized NCAP agent (i.e. no training)?

**Limitations:**

Yes.

**Strengths And Weaknesses:**

# Strengths

1) This is a unique perspective. It's rare for work in systems neuroscience to be fully translated into models that can be optimized with methods designed for deep learning.

2) The writing is clear throughout, though more insight into the motivation behind the unit types would be nice. E.g. way are oscillator units preferable to under ways of maintaining internal state (such as LSTMs)?

3) Ablations are thorough, though again more insight into these results would be appreciated.

# Weaknesses

1) It's unclear the extent to which learning matters here. Initial performance looks to be around 800 and final performance is perhaps 850. This is a weakness, because simply hardcoding a solution to a simple locomotion problem is not a significant AI result. I'm not convinced that this'll be useful for anything other than the exact task/body setup you use here.

2) The experimental results are very limited in scope. All of the AI papers mentioned in related works involve techniques being applied across a range of embodiments and tasks. And even within the constraints of the swimmer environment, none of these related works are compared against.

3) Misc. Figure 4 shouldn't use a log x axis (or at least have both versions visible somewhere). Transfer is stated as not possible for MLPs, but ignores work adapting deep learning methods to due just that: "One policy to control them all: Shared modular policies for agent-agnostic control". I'm not convinced that parameter-count is a meaningful metric in this regime, but even if it were "Weight Agnostic Neural Networks" could likely produce a single-parameter policy class with similar performance properties to NCAP.

#Takeaway

I would strongly encourage resubmitting after more general results are obtained. Alternatively, I think this work could be reframed as a contribution to neuroscience rather than AI, by emphasizing the testing the functional role of various nematode cells, etc and downplaying the potential utility in AI applications. Though this would still require considerable reworking and would likely be better served by a different publication venue.

---

> ### Author Response · Authors · 2022-08-02
> **Response to R2**
>
> Thank you for your feedback. We appreciate your highlighting the paper's unique perspective, clear writing, and thorough ablations. Here are our responses to your questions and concerns:
>
> **Pure learning vs priors: "It's unclear the extent to which learning matters here. Initial performance looks to be around 800 and final performance is perhaps 850. This is a weakness …"**
>
> We respectfully disagree: While the difference that learning makes here is admittedly limited, we don't see this as a weakness per se — in fact, it shows that our "prior" is working correctly. In works featuring behavioral priors [1,2], a tabula-rasa architecture (e.g. MLP) is trained to imitate demonstrations, then learning is slowed or frozen, and a higher-level policy can control the acquired movement primitives. In works featuring trajectory priors [3], the movement primitives are "hardcoded" through equations of motion. In our work featuring architectural priors, movement primitives are initialized and constrained through ANN structure. Much of this structure is indeed "hardcoded" with limited plasticity for finetuning. This mirrors the mechanisms of biology: we discuss the worm's circuits in the paper, but quadruped and even human locomotion circuits exhibit a priori structure [4] (honed through evolution) with limited plasticity constrained by the circuit composition. The idea of porting such structure into AI motor control settings is the primary contribution of this work, and we do not think that the limited learning in worm circuits affects that. Engineering prior knowledge/structure into ANNs is important for reinforcement learning and robotics (**R1**), even if it runs counter to a pure learning approach.
>
> **Scalability to other body/tasks: "Would NCAP work for non-Swimmer environments?"**
>
> The Swimmer NCAP example presented here would be useful for worm-like bodies. However, the broader concept of a neural circuit architectural prior is, in principle, applicable to a wide class of bodies. This is because they reflect the actual working solutions used by biological organisms, which have long set the performance standards for AI motor control. Please kindly refer to our response to **R1** about how a Quadruped NCAP could take a similar approach as Swimmer NCAP.
>
> **Parameter counts: "Why should we care about the parameter counts of small models independent of data efficiency?"**
>
> That's a good question. We agree that even the largest model considered here isn't bottlenecked by memory or compute. So as you suggest, there may no real practical reason to care (at least, for this simple case). But perhaps there's a more philosophical reason: Is it satisfying that ~73,000 parameters are needed to learn even this simple swimming task with MLPs? As we show in Figure 4B, MLP performance dramatically deteriorates as the number of parameters is reduced, yet even the smallest MLP tested has 74 parameters, which is an order of magnitude more than the 4 parameters in the Swimmer NCAP. These results suggest that Swimmer NCAP performs well not merely because it has fewer parameters than MLPs, but rather because it captures useful structure about the problem. The reward/params metric mainly serves to underscore this point: that a parameter in the NCAP architecture is in a sense "worth more" than one in an MLP. Why don't small MLPs learn via PPO/DDPG/ES for this simple task? What advantages can smarter ANN structure similarly have for more complex tasks? These are questions that future work may care to investigate.
>
> **Transfer results: "Transfer is stated as not possible for MLPs, but ignores work adapting deep learning methods to do just that"**
>
> We should clarify what we meant: We were referring the kind of trivial transfer that exploits modularity of a network architecture. For example, a convolutional neural network, by exploiting its spatial locality, can trivially be applied to images larger or smaller than it was trained on (within limits). Similarly, the Swimmer NCAP, by exploiting its segmented structure, can trivially be applied to bodies longer or shorter than it was trained on (within limits). This type of transfer is not so trivial to do with an MLP, with weight matrices constrained by input/output dimensions. We are familiar with the works you refer to, both of which also decompose their policy in a modular way — through a computational graph [4] or network of message passing modules [5]; the resulting architectures are therefore not MLPs.
>
> [1] "Neural probabilistic motor primitives for humanoid control", Merel et al., 2019.
>
> [2] "Learning agile robotic locomotion skills by imitating animals", Bin Peng et al., 2020.
>
> [3]  "Controlling legs from locomotion — insights from robotics and neurobiology", Buschmann et al. 2015.
>
> [4] "Weight agnostic neural networks", Gaier & Ha, 2019.
>
> [5] "One policy to control them all", Huang et al., 2020.

---

> > ### Comment · Reviewer_QpQM · 2022-08-08
> > **Significance of learning**
> >
> > > We respectfully disagree: While the difference that learning makes here is admittedly limited, we don't see this as a weakness per se — in fact, it shows that our "prior" is working correctly. In works featuring behavioral priors [1,2], a tabula-rasa architecture (e.g. MLP) is trained to imitate demonstrations, then learning is slowed or frozen, and a higher-level policy can control the acquired movement primitives. In works featuring trajectory priors [3], the movement primitives are "hardcoded" through equations of motion. In our work featuring architectural priors, movement primitives are initialized and constrained through ANN structure. Much of this structure is indeed "hardcoded" with limited plasticity for finetuning. This mirrors the mechanisms of biology: we discuss the worm's circuits in the paper, but quadruped and even human locomotion circuits exhibit a priori structure [4] (honed through evolution) with limited plasticity constrained by the circuit composition. The idea of porting such structure into AI motor control settings is the primary contribution of this work, and we do not think that the limited learning in worm circuits affects that. Engineering prior knowledge/structure into ANNs is important for reinforcement learning and robotics (R1), even if it runs counter to a pure learning approach.
> >
> > This would be fine if the motivation for this work was to provide a hand-crafted behavior space for e.g. down-stream hierarchical control, but the paper and experiments aren't written from this perspective. Rather, the primary motivation appears to be that this behavior prior *allows for subsequent learning*. If your experiments only show a ~5% performance increase during training, then that casts significant doubt on the extent of subsequent learning that is possible.
> >
> > Going back to one of my unanswered questions, why not show experiments with different reward functions so there's still room for learning? Going to a goal location is a particularly simple example: locomotion is necessary, but performance should still be initially low since the goal location isn't part of the behavioral prior. This could showcase your method's ability to learn without significant additional effort.

---

> > > ### Author Response · Authors · 2022-08-09
> > > **Navigate task to showcase learning**
> > >
> > > **Navigate task: "Why not show experiments with different reward functions so there's still room for learning? Going to a goal location is a particularly simple example: locomotion is necessary, but performance should still be initially low since the goal location isn't part of the behavioral prior. This could showcase your method's ability to learn without significant additional effort."**
> > >
> > > Thanks, this is a great suggestion for how to showcase the effect of learning beyond initial performance. We recently tested this out in preliminary experiments (not reported in paper) where we train on such a "navigate" task (the standardized variant in DeepMind Control Suite). We used a lower-level NCAP-Swimmer motor module, controlled by a higher-level MLP task module. As you expected, performance started low then improved as the MLP task module learned. Significantly, algorithms like PPO and OpenAI-ES *weren't able to learn at all using a generic MLP* (i.e. a flat learning curve), presumably because the navigate task provides quasi-sparse rewards that makes it difficult to learn to swim while learning to go to a goal; this is in line with previously reported RL algorithm benchmarks [1] (see "swimmer-swimmer6" on pg. 11). In contrast, our architectural prior facilitated better exploration at the start and contributed to faster learning of this hard task.
> > >
> > > Unfortunately, given the limited time left in this rebuttal period, we're unable refactor the manuscript to present these results formally; however, we'd be keen to include them in the camera-ready version!
> > >
> > > [1] "Tonic: A deep reinforcement learning library for fast prototyping and benchmarking", Pardo 2021.

---

> > > > ### Comment · Reviewer_QpQM · 2022-08-09
> > > > **Thanks for the update**
> > > >
> > > > Thank you for trying out the requested experiment. I understand these are tight timelines, but I'm afraid I can't significantly vote for acceptance based on the assumption that very significant changes will be made between now and the camera-ready version. That said, I'll still increase my score (3-->4) to represent the optimism I feel about a potential future version of this work.

---

### Official Review · Reviewer_R6Ad · 2022-07-12

**Rating:** 6
**Confidence:** 4
**Soundness:** 4 excellent
**Presentation:** 4 excellent
**Contribution:** 3 good

**Summary:**

The paper presents a method to embed priors for motor functions to artificial neural networks. The authors demonstrate a rather simple example of using architecture of C.elegans to control swimmer in mujoco control suite problem. The proposed method has similarities with previous methods such as CPGs, but it has clear differences as the authors explain. Using the architecture proposed by the authors, the controller for the 5 link swimmer is reduced to 4 parameters. The training results show pretty much instant learning curve with converged policies comparable to MLPs trained with Reinforcement Learning or Evolutionary Strategies.

**Questions:**

Would it be possible to give some guidelines on how to apply the given design principles to more complex problems such as legged locomotion (or even 2D walker problem)?

**Limitations:**

The main limitation is the lack of scalability of the method on more complex problems. It would be great to see authors give different architectures for different problems instead of a single one (swimmer).

**Strengths And Weaknesses:**

The main strength of the paper is to add prior knowledge and structure to artificial neural networks for motor controlling tasks. This is a very important point for reinforcement learning and robotics. It reduces the number of samples required for learning and it introduces interpretability to the network.

As one would expect, the results show that the same task can be learned with way less parameters in much smaller number of samples. Adding structure also provides some bounds to safety during learning, but the authors did not emphasize this point much.

The paper is very well written. The concepts and similar works are explained in details in a nice flow. It is very easy to follow the main idea while also understanding the details of the method.

In terms of weaknesses, as the authors acknowledge the authors demonstrate the method on a rather simple problem (swimmer), and they do not give a clear guideline to extend it to more complex motor control problems such as legged locomotion.

---

> ### Author Response · Authors · 2022-08-02
> **Response to R1**
>
> Thank you for your feedback and support! We hope these responses satisfactorily answer your questions:
>
> **Scalability to complex bodies: "Would it be possible to give some guidelines on how to apply the given design principles to more complex problems such as legged locomotion?"**
>
> Yes, for example: Quadrupeds (e.g. dog, cat, horse, rat) are a common class of bodies in both biology and robotics. The neural circuits controlling gait (walk, bound, gallop) are contained within the spinal cord and act semi-autonomously — producing robust locomotion even when top-down connections from the brain are lesioned [1]. Significant work in the neuromechanical modeling community has focused on quadruped locomotion [2, 3]. These works could be translated to AI motor control bodies as we do in this paper, and a Quadruped NCAP architecture would make use of similar integrators, oscillators, and constrained excitatory/inhibitory connections as Swimmer NCAP. Indeed, quadruped locomotion is something we are pursuing, but it outside the scope of this present conference paper. We see this paper as a case study that uses a simple organism to demonstrate several advantages of ANN architecture inspired by systems neuroscience and to motivate future work bridging the neuromechanical modeling and AI motor control communities.
>
> **Safety: "Adding structure also provides some bounds to safety during learning"**
>
> This is a great point. Simpler, sparser architectural structures may indeed be safer: their interpretability facilitates debugging and more constraints can be engineered. We've added this to the manuscript!
>
> [1] "Controlling legs from locomotion — insights from robotics and neurobiology", Buschmann et al. 2015.
>
> [2] "Computational modeling of spinal circuits controlling limb coordination and gaits in quadrupeds", Danner et al., 2017.
>
> [3] "Organization of the mammalian locomotor CPG", Rybak et al., 2015.

---

### Meta-Review · Area_Chair_Giup · 2022-08-23

**Recommendation:** Accept
**Confidence:** Less certain

**Metareview:**

This paper introduces the use of neural circuit architectural priors to build controllers for a physically simulated c-elegans-like swimmer implemented in MuJoCo as part of the DeepMind control suite. By leveraging the bio-inspired architectural priors, the controller starts with structured behavior (rather than highly erratic random movements as is commonly the starting point for embodied RL initial behavior). And the architectural prior supports continued learning from this starting point.

The work is seen as original, interesting, and quite clear. The work is also nicely self-contained.

That said, this paper has received mixed and borderline reviews (6, 4, 4, 6), and there were some concerns about scalability and utility to the AI community.  This paper was discussed with the SAC, and we decided that despite some of these legitimate concerns, this paper should be accepted. This paper has clear goals and can help us rethink some of our approaches to architectures.  Moreover the potential audience spans both neuroscience and AI.

We (the AC and SAC) still highly encourage you to seriously consider comments from the reviewers. From both the positive-leaning and negative-leaning reviewers, there is respect for what was done as a work of modeling, but concerns about whether this constitutes only well-done computational modeling, or if it really amounts to anything that could be useful for AI more generally (and if it could scale to other bodies). You outlined some next steps, and how similar approaches could be used in other scenarios and with more complex bodies; we recommend that you include that discussion in this paper. We'd also strongly encourage you to avoid assertive claims about how neuroscience-inspired ideas can generally improve AI systems, and acknowledge limitations in this case study. While this case study is a provocative first step, the reviewers and AC tend to believe it will prove quite difficult to extend this strategy to more complex bodies.

Overall, focusing on what was achieved in this paper, nice work.

**Award:**

No

---

### Decision · Program_Chairs · 2022-09-14

Accept